

**A database of paleoceanographic sediment cores from the North Pacific, 1951-2016**
Marisa Borreggine[1], Sarah E. Myhre[1,2*], K. Allison S. Mislan[1, 3], Curtis Deutsch[1], Catherine V. Davis[4]
[1] School of Oceanography, University of Washington, 1503 NE Boat Street, Box 357940, Seattle, WA

5     98195-7940

[2] Future of Ice Initiative, University of Washington, Johnson Hall, Room 377A, Box 351360, Seattle, WA

7     98195-1360

[3]eScience Institute, University of Washington, 3910 15th Ave NE, Box 351570, Seattle, WA 98195
[4]School of Earth, Oceans, and the Environment, University of South Carolina, 701 Sumter Street, Earth &
Water Science Building, Room 617, Columbia, SC 29208
*Corresponding author



**Abstract**
We assessed sediment coring, data acquisition, and publications from the North Pacific (north of 30˚N)
from 1951-2016. There are 2134 sediment cores collected by American, French, Japanese, Russian, and
international research vessels across the North Pacific (including the Pacific Subarctic Gyre, Alaskan
Gyre, Japan Margin, and California Margin, 1391 cores), Sea of Okhotsk (271 cores), Bering Sea (123
cores), and Sea of Japan (349 cores) reported here. All existing metadata associated with these sediment
cores are documented, including coring date, location, core number, cruise number, water depth, vessel
metadata, and coring technology. North Pacific age models are based on isotope stratigraphy, radiocarbon
dating, magnetostratigraphy, biostratigraphy, tephrochronology, % opal, color, and lithophysical proxies.
Here, we evaluate the iterative generation of each published age model and provide documentation of
each dating technique used, as well as sedimentation rates and age ranges. We categorized cores
according to availability of a variety of proxy evidence, including biological (e.g. benthic and planktonic
foraminifera assemblages), geochemical (e.g. heavy metal concentrations), isotopic (e.g. bulk sediment
nitrogen and carbon isotopes), and stratigraphic (e.g. preserved laminations) proxies. This database is a
unique resource to the paleoceanographic and paleoclimate communities, and provides cohesive
accessibility to sedimentary sequences, age model development, and proxies. The data set is publicly
available through PANGAEA at doi:https://doi.pangaea.de/10.1594/PANGAEA.875998.











## 1 Introduction

Paleoceanographic sediments provide the sedimentary, geochemical, and biological evidence of past earth system changes, and are one of the primary ways to investigate past changes in global and regional climate, ocean circulation, volcanism, and biogeochemical cycles, among many other ocean-related inquiries. Sediment cores are collected from the seafloor during oceanographic research cruises. After collection, sediment cores are processed, archived, analyzed, and the results are published in a scientific journal. Mechanistic hypotheses investigated by sediment core research include ocean-basin scale changes in deep ocean circulation (e.g. Rae et al., 2014; De Pol-Holz et al., 2006), deep-water and intermediate water formation and ventilation (e.g. Knudson and Ravelo, 2015a; Zheng et al., 2000; Cook et al., 2016), and changes in the oceanic preformed nutrient inventories (e.g. Jaccard and Galbraith, 2013; Knudson and Ravelo, 2015b), as well as more regional mechanisms such as sea ice extent (e.g. Max et al., 2012), upwelling intensity (e.g. Di Lorenzo et al., 2008; Hendy et al., 2004), local surface ocean productivity (e.g. Serno et al., 2014; Venti et al., 2017), and terrigenous and marine fluxes of iron (e.g. Davies et al., 2011; Praetorius et al., 2015).

When taken together, suites of cores can create robust reconstructions of large oceanographic provinces, and provide insight into earth system mechanistic hypotheses. However, there is not a common repository for paleoceanographic data and publications, and this lack of centralization limits the efficacy of the earth science community to direct research efforts. Paleoceanographers have benefited from the use of large databases of climate data in the past, such as CLIMAP (Climate: Long Range Investigation, Mapping, Prediction), which produced globally resolved temperature records for the Last Glacial Maximum (LGM) and aimed to determine climate system sensitivity from paleoclimate reconstructions (Hoffert & Covey 1992). The PaleoDeepDive project employed a similar approach to the systematic extraction of archival data and constructed a new way to assess and engage with paleobiological data (Zhang, 2015). These projects are examples of international research teams approaching the same limit—extraction and organization of dark data—that arises when creating comprehensive paleo-reconstructions.




## 1.1 Assembling a paleoceanographic database

There is a clear need to generate high quality paleo-environmental reconstructions and fit the North
Pacific into the global paleoceanographic framework, because the role this enormous ocean basin plays in
earth system changes remains unclear relative to its Southern Ocean and North Atlantic counterparts. We
present here a comprehensive database of 2134 sediment cores which may be utilized to generate a
broader context for the history of large oceanographic provinces as well as contained regional processes.
Here we describe this new database and the broad-scale findings of our census of coring metadata, age
model development, and proxy publications. We address the following questions in this manuscript to
supplement and provide context for our database:
1. Where have sediment cores been extracted from the North Pacific (North of 30˚N, including

the Pacific Subarctic Gyre, Alaskan Gyre, Japan Margin, and California Margin), the Bering

Sea, the Sea of Okhotsk, and the Sea of Japan? What metadata is available—regarding core

name, recovery date, recovery vessel and scientific agency, latitude and longitude, water

depth, core length, and coring technology in published cruise reports or peer-reviewed

investigations?

2. For sediment cores with published age models—what lines of evidence were used to develop

the chronological age of the sediment, what is the age range from core top to core bottom,

and what are the sedimentation rates?

3. What lines of sedimentary, geochemical, isotopic, and biological proxy evidence are

available for each sediment core?

4. What is the state of sediment core research effort and metadata reporting since the beginning

of paleoceanographic core research?


## 1.2 Age model development, paleoceanographic proxies, and cruise-core nomenclature,



Marine sedimentary age models tie the sedimentary depth (in meters below sediment surface) to calendar
age (ka, thousands of years), and the dating techniques used to complete these chronologies depend on the
quality, preservation, and age of the sediments, as well as the investigative priorities of research teams
and the proximity of other well-developed sedimentary chronologies. Not all sedimentary chronologies
are of the same quality. Building a database with all dating techniques parsed and age models iterations
captured, along with reported sedimentation rates and the sedimentary age ranges, will provide
investigators with the capacity to quickly evaluate the specific cores that meet the investigative priorities.
In turn, paleoceanographic proxies are a diverse suite of biological, isotopic, geochemical, and
sedimentary observations and measurements taken from sediments. A catalogue of paleoceanographic
proxies, and associated publications, provides an efficient resource for assessing the availability and
quality of many different lines of evidence.

Sediment cores are often represented by their cruise-core unique identifier, which has the general format
"cruise name-core number". The cruise number is generally indicative of the research vessel employed
and the year of the expedition, for example, YK07-12 is the 12th cruise of R/V Yokosuka in 2007. Cruise
L13-81 is the 13[th] cruise of S.P. Lee in 1981, and MR06-04 is the 4[th] cruise of R/V Mirai in 2006. Often
the core number will be preceded by a PC (piston core), MC (multicore), TC (trigger core), or GC
(gravity core) to signify the coring technology. The sediment core B37-04G is the 4[th] gravity core from
the 37[th] cruise of R/V Professor Bogorov, and EW9504-11PC is the 11[th] piston core from the fourth cruise
of R/V Maurice Ewing in 1995. However, this nomenclature is not comprehensive, and for example cores
affiliated with iterations of the International Ocean Discovery Program are represented by the program
abbreviation and their hole number (i.e. ODP Hole 1209A). The cruise name or number is unknown for
many cores, and in these cases, the core is referred to by its number.

**2 Methods**



Here we assemble a database from peer-reviewed publications, publicly available online cruise reports,
and print-only cruise reports available through the University of Washington library network. Detailed
metadata is reported for cores where it is available, commonly from cruise reports, including water depth
(in meters), recovery year, latitude and longitude, coring technology, and core length (in meters).
Summary details regarding affiliated research vessels and institutions were gathered from publications or
cruise reports. Here we catalog the methodological approaches taken to develop age models in the North
Pacific by tracking the sequential publications to capture each iteration and line of evidence used in the
most up-to-date published version. We reported core top and bottom ages, along with sedimentation rates.
For each core, we documented published proxy evidence, including isotopes stratigraphy, geochemistry,
micropaleontology, and sediment analyses.

**3.0    Results**
**3.1    Sediment coring and metadata**
We documented 2134 sediment cores and 283 marine geology research cruises above 30°N, from 1951 to
2016, in the North Pacific, the Bering Sea, the Sea of Japan, and the Sea of Okhotsk (Figure 1, Table 1).
The majority of sediment cores were extracted from the Northern Subarctic Pacific (1391 cores), followed
by the Sea of Japan (349 cores), the Sea of Okhotsk (271 cores), and the Bering Sea (123 cores).  Many of
the oldest cores in this oceanographic province come from the central abyssal Pacific and are associated
with cruises of the Deep Sea Drilling Project. The recovery ages of cores range from 1951- 2010 (Figure
1, Table 1). Frequently, metadata associated with sediment cores or marine research cruises are
unavailable to the public or omitted from publications affiliated with sediment cores. For example, 495
cores are in the literature without recovery year, 354 sediment cores were published without latitude and
longitude, and 642 cores were reported without specifying the coring technology used (Table 1, 2). Even
more, 1210 ancillary sediment cores reported in our database were identified in supplemental tables
within publications, however no original cruise reports or peer-reviewed publications otherwise report on
these cores.




## 3.2 Sediment chronologies

In the North Pacific, 519 marine sediment cores have published age models, and 266 of these

chronologies are generated with radiocarbon dating ($^{14}$C) of materials such as planktonic foraminifera,

molluscs, and terrigenous material like bark or wood fragments (Figure 2). Radiocarbon dating is the

most common chronological technique region-wide (51% of age models incorporate this method). Lead

dating ($^{210}$Pb) is used for 12 sediment chronologies. Many other lines of evidence are used in the North

Pacific and marginal seas to develop paleoceanographic age models. These approaches vary regionally

and include planktonic foraminifera oxygen isotope stratigraphy, diatom silica oxygen isotope

stratigraphy, biostratigraphy, magnetostratigraphy, tephrochronology, chronostratigraphy, carbonate

stratigraphy, opal stratigraphy, composition, lithophysical proxies, the presence of lamination, chlorin

content, and color (a*, b*, and L* values) (Figure 2, Table 3). For example, in the Sea of Japan,

lithophysical proxies such as core laminations are often utilized as chronological proxy evidence, and

12% of local core age models incorporate this technique. Tephrochronology is also applied in 51% of Sea

of Japan age models due to regional volcanism. In the Bering Sea, peaks in silica are often used, and 13%

of the regional age models incorporated this technique. Published sedimentation rates ranged across the

North Pacific (0.1-2000 cm/ka), Bering Sea (3-250 cm/ka), Sea of Okhotsk (0.7-250 cm/ka), and the Sea

of Japan (0.2-74 cm/ka), with the highest rates within the Alaska Current in the North Pacific (up to 2000

cm/ka).

## 3.3 Paleoceanographic proxies from marine sediment cores

From all reported sediment cores in the North Pacific and marginal seas, 40% of cores have published

proxy data (Figure 3, 4). Stable isotope stratigraphy is available for 293 cores, including oxygen, carbon,

and nitrogen isotopes ($\delta^{18}$O, $\delta^{13}$C, $\delta^{15}$N) from planktonic and benthic foraminifera. We documented 236

cores with planktonic foraminifera oxygen isotopes, 178 cores with benthic foraminifera oxygen isotopes,

67 cores with planktonic foraminifera carbon isotopes, 77 cores with benthic foraminifera carbon



isotopes, and 34 cores with bulk sediment nitrogen isotopes (Table 4). The lines of proxy evidence
documented in the database address large thematic questions in paleoceanography, including ocean
temperature, paleo-biology, seafloor geochemistry, sea ice distribution, and additional sedimentary
analyses (Figure 3, Table 4).

We recorded paleothermometry data for 234 cores. Sea surface temperature is reconstructed using
planktonic foraminifera oxygen isotopes ($\delta^{18}O_p$), magnesium/calcium ratios from planktonic foraminifera,
and alkenones, including $TEX_{86}$, and $U^k_{37}$ (Figure 3). We recorded 425 cores with biostratigraphy and
assemblage abundance data for foraminifera, diatom, radiolarian, ostracod, silicoflagellate, ebridian,
acritarch, coccolithophore, and dinoflagellate assemblages (Figure 3, Table 4). Biostratigraphy utilizes
known microfossil assemblages and their corresponding ages to assign a geologic date to core strata
containing assemblages.  Geochemical analysis is reported for 151 cores (Figure 3, Table 3).
Geochemical data comes from measurements of, for example, brassicasterol, magnesium, calcium,
molybdenum, cadmium, manganese, uranium, chromium, rhenium, chlorin, titanium, iron, zinc, and
beryllium. We documented 234 cores with sea ice proxy data (Figure 3). Sea ice proxies include
geochemical biomarker IP25, ice-rafted debris (IRD), and diatom communities exclusive to sea ice
environments. The presence and concentration of these proxies are indicative of contemporary sea ice
extent and volume. We recorded 521 sediment cores with lithophysical analyses, including the
documentation of core lamina, sediment density, mass accumulation rates, biogenic opal and barium,
silicon/aluminum weight ratios, carbon/nitrogen ratios, inorganic and organic calcium carbonate and
carbon, inorganic nitrogen, sulfur, and clay mineral composition (Figure 3).

**3.4    Research cruise and publication rate**
We cataloged 565 peer-reviewed publications and cruise reports and evaluated the progress of
paleoceanographic research using a suite of annual assessments of cruise and core metadata and
publications. We evaluated the annual number of age models published (using any dating technique), age





models published specifically with radiocarbon dating, publications generated, sediment cores collected,
research cruises completed, and the mean number of proxies generated per core (Figure 4). Cruise reports
were not publicly available for every cruise or core, and many cores cited in cruise reports were never
published upon in peer-reviewed literature. The number of affiliated publications and cruise reports per
core ranged from 0 to 23 publications (Figure 5). Only 41 cores have more than 6 publications, while the
majority of cores (1210 cores, or 57% of all cores extracted from the North Pacific) lack any publication.

The state of North Pacific paleoceanographic investigations has evolved incrementally in the 65-year
history of research in the region, and we characterize the history into two distinct phases (before and after
the early 1990s). Core recovery rates were high from 1951-1988, a period when expeditions were driven
by individual institutions, wherein peer-reviewed publication was not the primary research outcome and
therefore publication rates were low (Figure 4). Annual rates of cruise completion and sediment core
extraction peaked in 1965-1968, and this is a consequence of the temporal overlap in peak research efforts
by Scripps Institute of Oceanography (1951-1988), Lamont-Doherty Earth Observatory (1964-1975),
Oregon State University (1962-1972), and the Deep Sea Drilling Project (1971-1982).

Annual rates of publication (peer-reviewed and cruise reports), including those publications with age
models, increased around 1995 (Figure 4). In this later period, research cruise efforts were dominated by
international research team efforts and resulted in increased peer-reviewed publications, sediment core
chronology constructions, and proliferation of radiocarbon dating. There are 41 cores with the highest
level of documentation (>6 publications), and these archives are primarily located within the California
Current (Figure 5). Major peaks in cruising and coring efforts coincided with research cruises by the
International Ocean Discovery Program, such as ODP Legs 145 and 146 in 1992 (North Pacific Transect,
Cascadia Margin), ODP Leg 167 in 1996 (California Margin), and IODP Expedition 323 in 2009 (Bering
Sea). Despite the increase in publications around 1995, we observe no distinct temporal trend in the rate
of cruise completion and coring effort (Figure 4).




## 4    Discussion

### 4.1    Evolving state of North Pacific coring and paleoceanographic approaches

Extensive cruise and research efforts have focused on the marine geology of the North Pacific. Often, the
cruise and core metadata from these efforts is unpublished, though they are integral to collaboration,
continued research, and publication. Here, we present a database with 2134 sediment cores, 283 research
cruises, and 565 peer-reviewed publications related to paleoceanographic research (Table 1). We
cataloged 519 publications with sedimentary age models, and of those age models 266 utilize radiocarbon
dating. Throughout the North Pacific, Bering Sea, Sea of Okhotsk, and Sea of Japan, the techniques for
reconstructing sedimentary age models regionally varied. Our database encompasses all available lines of
proxy evidence for sea surface temperature reconstructions, paleobiological assemblages, seafloor
geochemistry, sea ice reconstruction, and other sedimentary analyses. We observe and discuss two
distinct periods of research and publication effort (before and after the early 90s) (Figure 4).

We documented a community-wide shift away from singular dating techniques toward age models which
incorporate several techniques. Multiproxy approaches hold merit through verifying or constraining the
results of a singular proxy, and thereby disentangling multiple environmental drivers and providing
redundancy in order to create records of climate and ocean conditions from sediment cores (Mix et al.,
2000; Mann, 2002). Publication count and age model development has increased through the last 60 years
and evolved from singular dating techniques to more detailed high-resolution age models constructed to
investigate millennial and submillennial paleoceanographic variability (Figure 4, 5).

### 4.2    The merit of database management and open-access science

Databases are integral to facilitate efficient, hypothesis-based investigations into earth system
mechanisms. Public access to databases facilitate a higher volume of research by a diverse range of
scientists (Harnad and Brody 2004). The necessity for databases to encompass a wide array of data over





large oceanographic provinces is also largely recognized. Open access tools from PANGAEA support
database-dependent research, because hypothesis-based investigations can be make more efficient
through public access. For example, content from online databases has contributed to research in
atmospheric forcing (e.g. Shaffer et al., 2009), Atlantic Meridional Overturning Circulation (AMOC, e.g.
Schmittner and Galbraith, 2008), and Southern Ocean ventilation (e.g. Yamamoto et al., 2015). The
metadata in these databases must be thorough, as data is impractical without the affiliated identifier,
location, and methods (Lehnert et al., 2000). Database management should be a priority in any field that
incorporates the contents of online repositories of knowledge and research. The disconnect between the
research goals of the paleoceanographic community and the metadata produced here can be described as a
"breakdown", a limit on the progress of paleoceanographic research (Tanweer et al., 2016). These
breakdowns allow us to self-assess and move forward with insight into best practices. Metadata is
produced from datasets that are inherently human in design, and therefore are not inerrant. Assessment of
the errors in metadata reporting can directly reveal the need for community-wide improvements. As an
example, all cores should be reported with latitude and longitude; the absence of this specific metadata
significantly impairs further work. The database presented here, as well as others like it, consolidates the
research effort of an entire community into an efficient tool for future investigative purposes.

**4.3     Recommendations for the marine geology community**
Ocean sediment records are one of the primary tools for understanding earth's history, and the
documentation of these records benefits the entire community of earth scientists. The publications and
cruise reports here represent a large body of research completed on North Pacific sediment records,
however this may not constitute the entire body of work. We demonstrate here a need for more thorough,
accessible documentation of marine geology and paleoceanographic research. In our examination of
publications, cruise reports, and notes from research cruises, we gained insights into past inconsistencies
in marine sediment record reporting. We recommend a suite actions to ensure efficient, comprehensive
sediment core collection, metadata documentation, and the publication of chronologies and proxies. We



propose that each publication thoroughly reports metadata on all cores discussed, and their associated
cruises, including core unique identifier numbers, cruise name and number, vessel name, geographic
coordinates, core recovery date, core length, core recovery water depth (meters below sea level), sampling
resolution, sampling volume, core archival repository, and the link (if existing) to public cruise reports.
We also suggest summarizing each core's metadata, age model, and publication history of previous
publications in the methods section in order to provide a frame of reference for new findings, especially in
the context of iterative age model revisions.

**5        Author Contribution**
S. E. Myhre and C. V. Davis initiated the building of this database. M. Borreggine and S. E. Myhre built
the database and were joined by K. A. S. Mislan in producing figures and analysis for this manuscript. All
authors wrote the manuscript.

**6        Acknowledgements**
The authors would like to thank the University of Washington Library Oceanography collection, namely
Louise Richards and Maureen Nolan. We also wish to acknowledge the support for this publication,
provided by the University of Washington Purple and Gold Scholarship, the UW School of Oceanography
Lowell K. and Alice M. Barger Endowed Scholarship, the Clarence H. Campbell Endowed Lauren
Donaldson Scholarship, the UW College of the Environment Student Travel Grant, and NSF Grant OCE-
1458967. K.A.S. was supported by the Washington Research Foundation Fund for Innovation in Data-
Intensive Discovery and the Moore/Sloan Data Science Environments Project at the University of
Washington. The database can be found online at (doi:X) from PANGAEA.






**Figure and Tables**

**Table 1**. Regional summary of sediment core database for the Bering Sea, North Pacific, Sea of Japan, and Sea of Okhotsk, including number of cores recovered, the regional percent (%) of cores reported with latitude and longitude, number of research cruises, total regional publication count, count of cores with no peer-reviewed publications or cruise report, the range of core recovery years, the regional percent (%) of cores reported with recovery years, and the range of recovered core water depths (meters below sea level).

| Region | Bering Sea | North Pacific | Sea of Japan | Sea of Okhotsk | Total |
|---|---|---|---|---|---|
| Count of cruises | 24 | 179 | 78 | 31 | 283 |
| Count of publications | 70 | 306 | 114 | 90 | 565 |
| Count of cores with no publications* or cruise report | 61 | 849 | 172 | 128 | 1210 |
| Number of cores recovered | 123 | 1391 | 349 | 271 | 2134 |
| Percent of regional cores (%) reported with latitude and longitude | 91.9 | 81.9 | 77.7 | 91.9 | 83.4 |
| Range of core recovery years | 1957-2009 | 1951-2009 | 1957-2010 | 1972-2009 | 1951-2010 |
| Percent of regional cores (%) reported with recovery year | 87 | 68.9 | 96.9 | 98.5 | 76.8 |
| Range of recovered core water depths (meters below sea level) | 33-3930 | 21-9585 | 129-5986 | 105-8180 | 21-9585 |

* These cores are listed in large data tables in auxiliary publications, but the original reporting is not publicly available.



**Table 2.** Percent of regional cores reported with coring technology, and the number of cores recovered in
the North Pacific and marginal seas by coring technology. Additional reported coring technologies
include the less often utilized Hydrostatic cores, Kasten cores, Asura cores, Pressure cores, and Trigger
cores.

| Region | Bering Sea | North Pacific | Sea of Japan | Sea of Okhotsk |
|---|---|---|---|---|
| % of regional cores reported with drilling technology | 75.6 | 72.1 | 60.2 | 68.6 |
| # of Piston cores | 33 | 476 | 177 | 67 |
| # of Gravity cores | 11 | 250 | 15 | 73 |
| # of Box cores | 2 | 78 | 1 | 1 |
| # of Riserless Drilling cores | 7 | 52 | 0 | 0 |
| # of Multicores | 33 | 39 | 1 | 39 |
| # of Phleger cores | 0 | 15 | 0 | 0 |
| # of Jumbo Piston cores | 4 | 10 | 0 | 0 |
















**Table 3.** Summary statistics on core chronology, including the number of cores with radiocarbon dating
($^{14}$C) and oxygen isotope stratigraphy of planktonic foraminifera ($\delta^{18}$O), as well as the regional mean core
top and bottom ages, and the number of cores with published sedimentation rate ranges and means.

| Region | Bering Sea | North Pacific | Sea of Japan | Sea of Okhotsk |
|---|---|---|---|---|
| $^{14}$C Dating | 28 | 158 | 38 | 42 |
| $\delta^{18}$O Stratigraphy, Planktonic foraminifera | 20 | 122 | 29 | 32 |
| Mean core top age (Calendar age, ka) | 6.0 ± 9.6 | 9.5 ± 61.6 | 5.9 ± 28.4 | 1.1 ± 2.3 |
| Mean core bottom age (Calendar age, ka) | 523.7 ± 1146.2 | 6210.8 ± 20250.7 | 996.1 ± 2602.2 | 153.1 ± 279.2 |
| Cores with sedimentation rate range | 28 | 97 | 20 | 18 |
| Cores with sedimentation rate mean | 7 | 68 | 9 | 16 |
















**Table 4.** Regional summary of isotopic, geochemical, biological, and sedimentary proxies. Benthic and
planktonic isotopic analysis is for all cores, including, but not limited to, isotope analysis used in core
chronology development.

| Region | Bering Sea | North Pacific | Sea of Japan | Sea of Okhotsk |
|---|---|---|---|---|
| $\delta^{18}O_b$ | 19 | 118 | 15 | 25 |
| $\delta^{18}O_p$ | 20 | 139 | 45 | 32 |
| $\delta^{13}C_b$ | 7 | 57 | 7 | 6 |
| $\delta^{13}C_p$ | 6 | 50 | 10 | 1 |
| $\delta^{15}N$ | 5 | 16 | 12 | 1 |
| Alkenone $U^{K'}_{37}$ | 7 | 64 | 15 | 12 |
| $TEX_{86}$ | 0 | 1 | 0 | 0 |
| Foraminiferal Biostratigraphy | 12 | 82 | 22 | 8 |
| Foraminiferal Abundance | 8 | 53 | 41 | 18 |
| Diatom Biostratigraphy | 35 | 87 | 49 | 25 |
| Geochemical Proxies (Mg, Mo, Cd, Mn, U, Cr, Re, Ca, T, Fe, Mg, Zn, Be, Chlorin) | 1 | 189 | 46 | 27 |
| % Opal | 21 | 38 | 20 | 20 |
| Total Inorganic Carbon (%) | 16 | 204 | 49 | 26 |
| Total Organic Carbon (%) | 16 | 142 | 62 | 26 |
















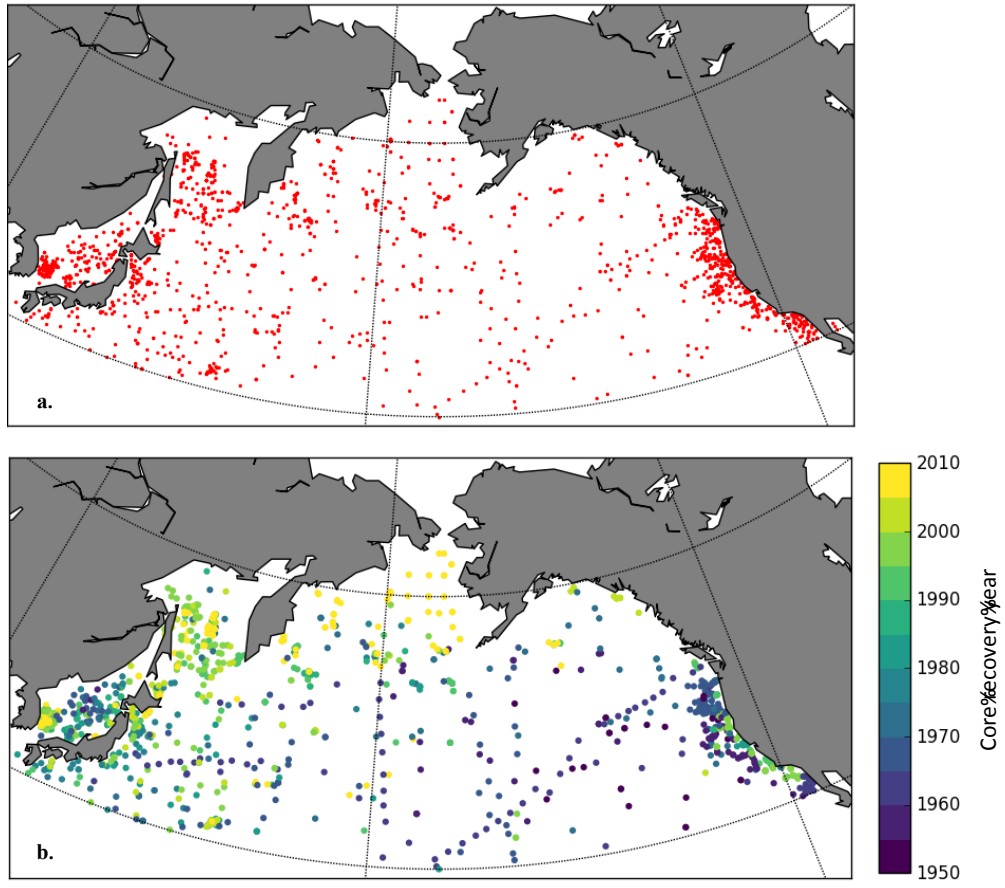


**Figure 1.** Location and recovery year of marine sediment cores from the North Pacific and marginal seas.

**a.** Sediment cores in the North Pacific (above 30°N) published latitudes and longitudes (354 additional

cores were documented in either cruise reports or peer-reviewed publications without latitude and

longitude). **b.** Sediment cores published with an associated core recovery year, ranging from 1951-2010,

and this age range corresponds with the color bar (495 cores have been published in either cruise reports

or peer-reviewed publications without recovery year).


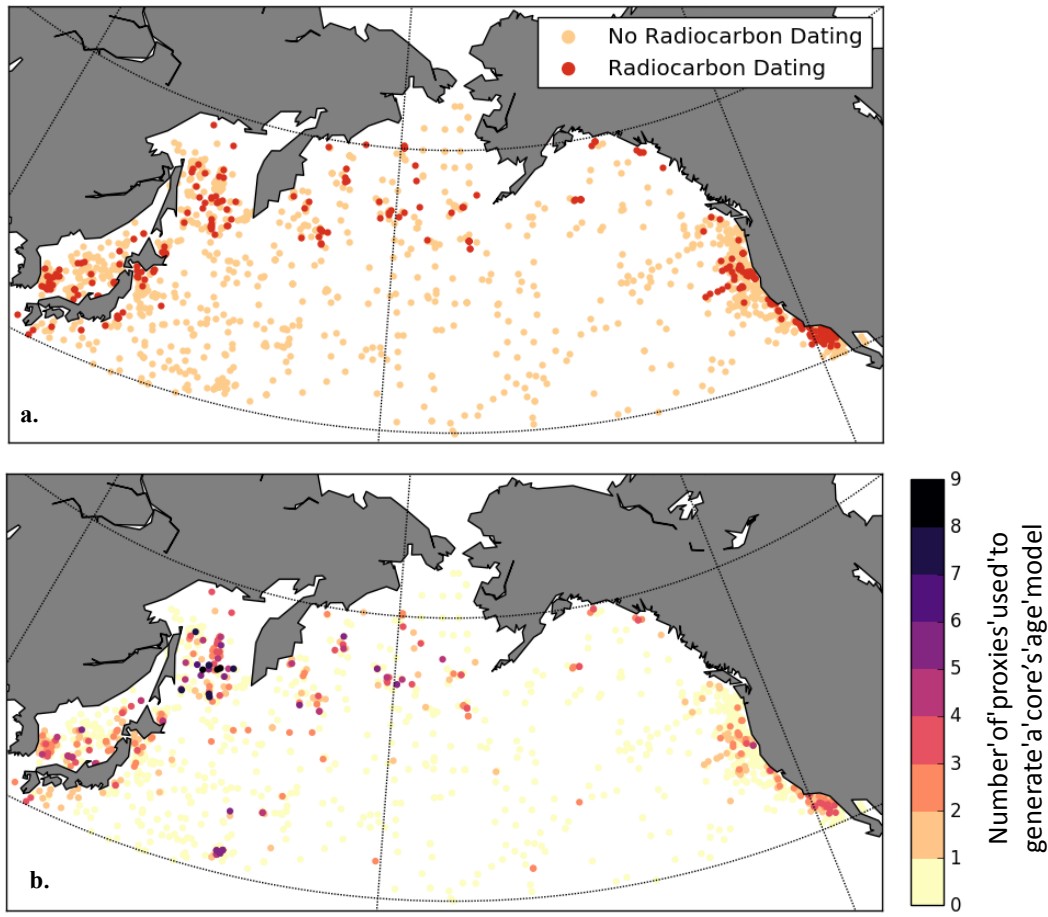


**Figure 2.** Age model development for sediment cores in the North Pacific and marginal seas. **a.** Cores

with age models that have been constructed with radiocarbon dating ($^{14}$C). **b.** Number of

paleoceanographic proxies used to generate each core's age model.








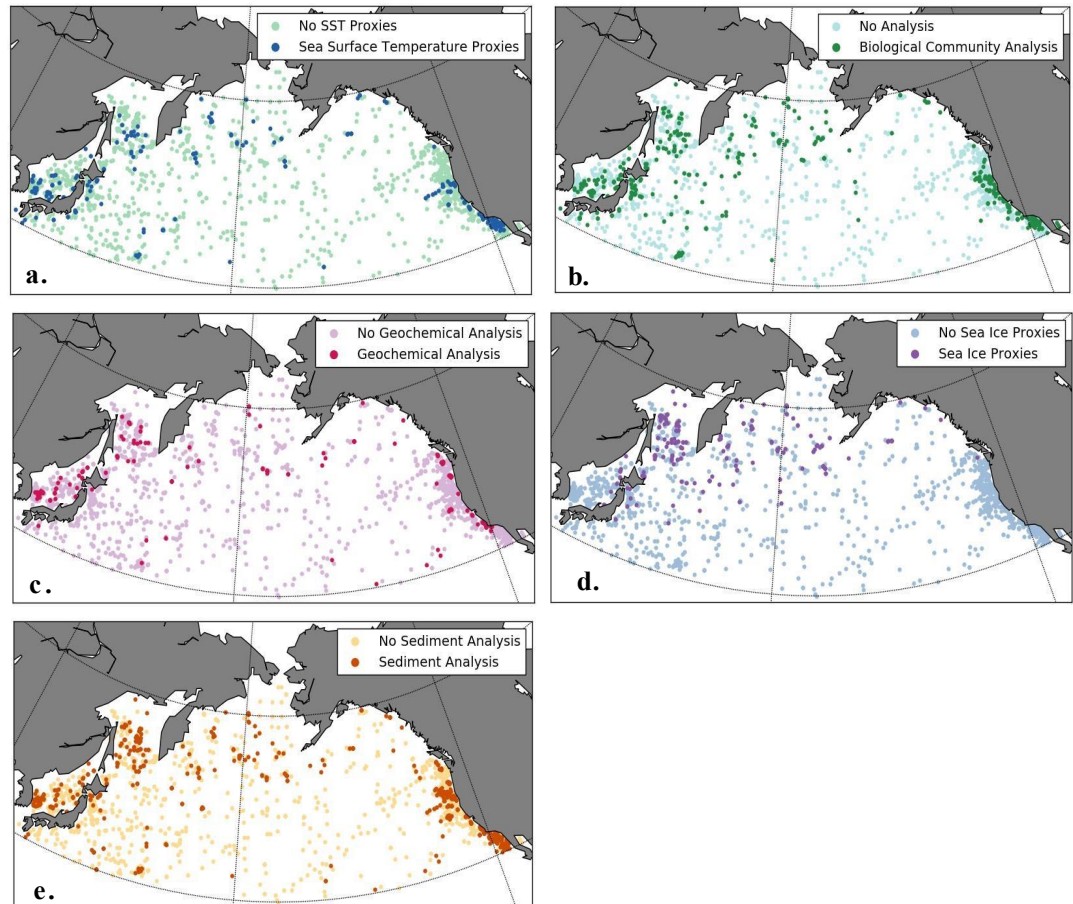


**Figure 3.** Published paleoceanographic proxies in the North Pacific. a. sea surface temperature

reconstructions including planktonic foraminifera oxygen isotopes ($\delta^{18}O_p$), magnesium/calcium

measurements, TEX$_{86}$, and U$^k_{37}$ alkenones b. biostratigraphy of microfossils, including foraminifera,

diatoms, radiolarians, ostracods, silicoflagellates, ebridians, acritarchs, coccolithophores, and

dinoflagellates **c.** geochemical proxy analysis, including trace metals such as brassicasterol, magnesium,

calcium, molybdenum, cadmium, manganese, uranium, chromium, rhenium, chlorin, titanium, iron, zinc,

and beryllium **d.** presence of sea ice proxies including geochemical biomarker IP25, ice-rafted debris

(IRD), and sea ice related diatom communities **e.** analysis of lithophysical core proxies, including

measurements of core lamina, biogenic opal and barium, silicon/aluminum weight ratio, sulfur, inorganic



and organic carbon weight and mass accumulation rates, mass accumulation rates of various elements,
inorganic nitrogen, carbon to nitrogen ratios, sediment density, and clay mineral composition.
















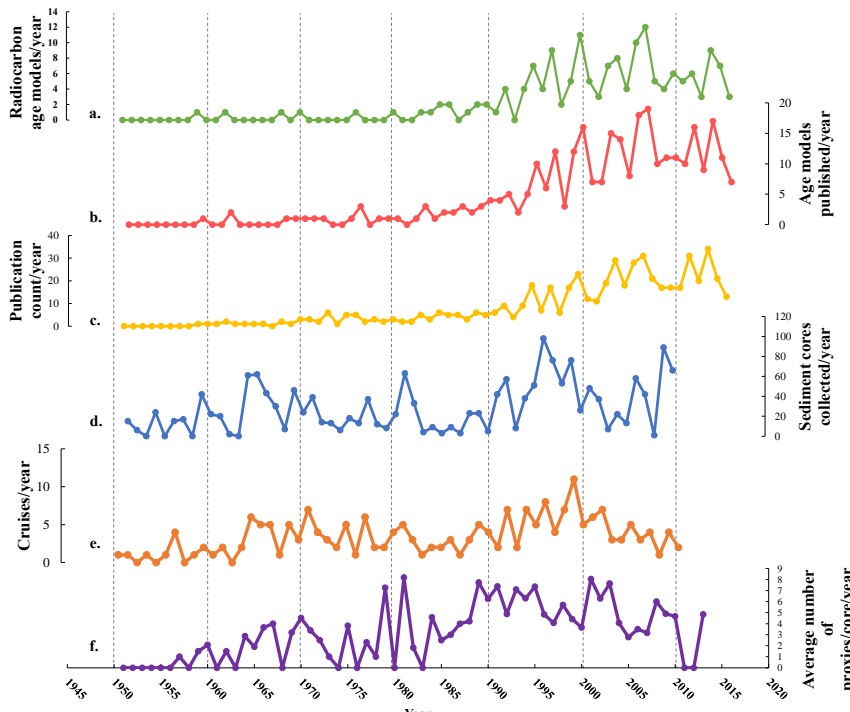


**Figure 4.** North Pacific and marginal seas marine geology cruise and paleoceanographic research

progress through time, wherein the lead-lag timing of cruise reporting and core publication is assumed for

the most recent years. We utilize peer-reviewed publications to locate cores, and there is a lag between

publication and core extraction. **a.** Annual number of age models published using radiocarbon dating. **b.**

Annual number of all age models published. **c.** Total annual number of publications and cruise reports. **d.**

Annual number of cores collected. **e.** Marine research expedition count per year. **f.** Annual mean number

of proxy analyses published per core.


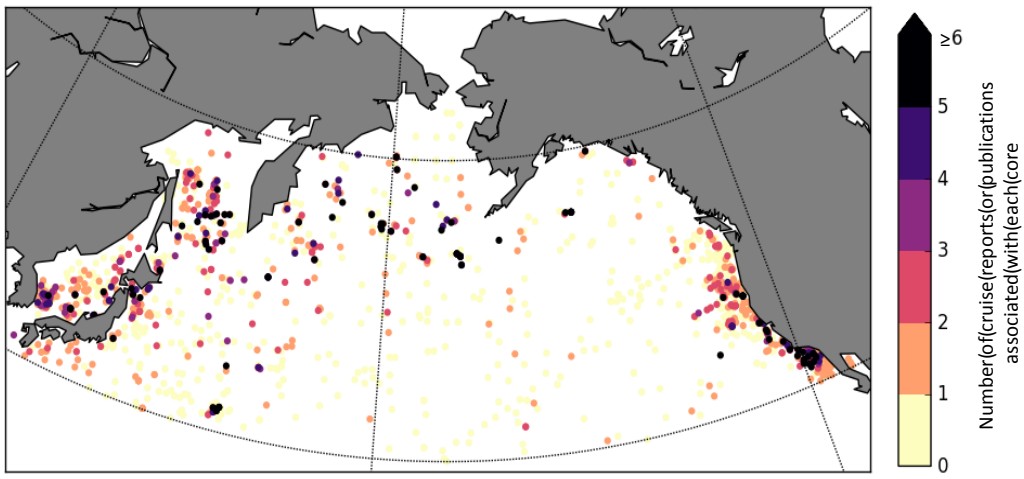

**Figure 5.** Number of affiliated publications and cruise reports for each core. Maximum publication count

for an individual core is 23.


















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
