# Peer review of "A database of paleoceanographic sediment cores from the North Pacific, 1951-2016"

_Earth System Science Data, 2017_

## Short Comment (SC1) · 22 Jun 2017

[revised manuscript text omitted]

---

## Referee Comment (RC1) · T. M. Cronin (Referee) · 7 Jul 2017

[referee-annotated manuscript omitted]

---

## Referee Comment (RC2) · B. Liu (Referee) · 20 Jul 2017

The authors presented a database of paleoceanographic sediment cores from the North Pacific from 1951-2016. Generally, the database is very useful for the scientific researches in marine geology and paleoclimate, etc. I suggest a few minor revisions. Several comments or suggestions are as follows: 1.Page 2 line, the word "heavy metal" should be replaced as "major or trace element", which is more common in paleoceanographic field. 2.Page 7, section 3.2. Sediment chronologies, many approaches are applied in sediment dating. I am wondering that if different age models are available in a same sediment core, which age model is more reliable. The paper needs discussion on the data quality comparison of different age models. 3.Page 7, section 3.3, It was also suggested that authors should try to provide a precision comparison discussion

between different sea surface temperature reconstruction approaches, especially for a sediment core, where different sea surface temperature reconstruction data are available. 4.The information on the sediment core length (e.g average length, minimum length and maximum length) should be provided in table 3. 5.Is Alkenone UK′37 in table 4 same as Uk37 in page 8, line 188 and page 19, line 392?

---

## Author Comment (AC1) · 25 Aug 2017

Thank you very much for your constructive comments and suggestions. Below we comprehensively address each critique and request.

"The authors presented a database of paleoceanographic sediment cores from the North Pacific from 1951-2016. Generally, the database is very useful for the scientific researches in marine geology and paleoclimate, etc. I suggest a few minor revisions. Several comments or suggestions are as follows: 1. Page 2 line, the word "heavy metal" should be replaced as "major or trace element", which is more common in paleoceanographic field."

We have changed this phrase to major trace element in line 39.

[Figure]

"2. Page 7, section 3.2. Sediment chronologies, many approaches are applied in sediment dating. I am wondering that if different age models are available in a same sediment core, which age model is more reliable. The paper needs discussion on the data quality comparison of different age models."

We did not observe alternate age models for individual sediment cores. Rather, we document the iterative refinement of age model through time and through additive publications. We extensively describe the data contributing to the process of age model generation, such that future investigators will be able to refine searches for age models by different quality standard. We do assert though that the qualification of data quality of different age models is outside the prevue of this manuscript – because that are many situationally-specific caveats toward understanding and interpreting age models.

3. Page 7, section 3.3, It was also suggested that authors should try to provide a precision comparison discussion between different sea surface temperature reconstruction approaches, especially for a sediment core, where different sea surface temperature reconstruction data are available."

We agree. This database will allow for extensive and precise comparisons of paleothermometry. However, these research avenues are currently outside of the prevue of this database, and will need to be perused in a subsequent investigation.

"4. The information on the sediment core length (e.g average length, minimum length and maximum length) should be provided in table 3."

Thank you for this suggestion. We have added sediment core length into table 3.

"5. Is Alkenone UK37 in table 4 same as Uk37 in page 8, line 188 and page 19, line 392?"

Yes, these are the same methods. We have corrected all terminology to UK37.

Please also note the supplement to this comment:

https://www.earth-syst-sci-data-discuss.net/essd-2017-49/essd-2017-49-AC1-supplement.pdf

[Figure]

**Supplement:**

**A database of paleoceanographic sediment cores from the North Pacific, 1951-2016**

Marisa Borreggine[1], Sarah E. Myhre[1,2*], K. A. S. Mislan[1, 3], Curtis Deutsch[1], Catherine V. Davis[4]

[revised manuscript text omitted]

**1.1    Assembling a paleoceanographic database**

A clear need exists for high quality paleo-environmental reconstructions to fit the North Pacific into a climate global framework, because the role this enormous ocean basin plays in earth system changes remains relatively unclear in comparison to the Southern and Atlantic Oceans. To address the collective need, we present here a new database of North Pacific paleoceanographic research efforts, along with the broad findings of our census of coring metadata, age model development, and proxy publications. We address the following questions in this manuscript to supplement and provide context for our database:

1.  Where have sediment cores been extracted from the North Pacific seafloor (North of 30°N,

  including the Pacific Subarctic Gyre, Alaskan Gyre, Japan Margin, and California Margin),

  the Bering Sea, the Sea of Okhotsk, and the Sea of Japan? What metadata is available in

  published cruise reports or peer-reviewed investigations, including core name, recovery date,

  recovery vessel and scientific agency, latitude and longitude, water depth, core length, and

  coring technology?

2.  For sediment cores with published age models, what lines of evidence were used to develop

  the chronological age of the sediment, what is the age range from core top to core bottom,

  and what are the sedimentation rates?

3.  What lines of sedimentary, geochemical, isotopic, and biological proxy evidence are

  available for each sediment core?

4.  How has the state of North Pacific research efforts and reporting changed since the beginning

  of paleoceanographic expeditions?

**1.2      Paleoceanographic age models, proxies, and nomenclature**

Marine sedimentary age models tie the sedimentary depth (in meters below sediment surface) to calendar age (ka, thousands of years and/or Ma, millions of years). Not all sedimentary chronologies are of the same quality, and often age models are iteratively refined. Age models are developed with many different dating techniques, which are dependent upon the quality, preservation, and age of the sediments, as well as the investigative priorities of research teams and the proximity of other well-developed sedimentary chronologies. Paleoceanographic proxies, including biological, isotopic, geochemical, and sedimentary observations and measurements, address large thematic questions in the reconstruction of ocean environments, including ocean temperature, paleobiology, seafloor geochemistry, sea ice distribution, and additional sedimentary analyses.

Sediment cores are often represented by their cruise-core unique identifier, which has the general format

"cruise name-core number". The cruise number is generally indicative of the research vessel employed and the year of the expedition. For example, L13-81 is the 13[th] cruise of S.P. Lee in 1981, MR06-04 is the

4[th] cruise of R/V Mirai in 2006, and YK07-12 is the 12th cruise of R/V Yokosuka in 2007. Often the core number will be preceded or followed by a PC (piston core), MC (multicore), TC (trigger core), or GC

(gravity core) to signify the coring technology. The sediment core B37-04G is the 4[th] gravity core from the 37[th] cruise of R/V Professor Bogorov, and EW9504-11PC is the 11[th] piston core from the fourth cruise of R/V Maurice Ewing in 1995. However, this nomenclature is not comprehensive. For example, cores affiliated with iterations of the International Ocean Discovery Program are represented by the program abbreviation and their hole number (i.e. ODP Hole 1209A). Cruise name or number is unknown for many cores, and in these cases the core is referred to by number.

**2 Methods**

Here we assembled a database from peer-reviewed publications, publicly available online cruise reports, and print-only cruise reports available through the University of Washington library network. Detailed metadata was reported for cores where it is available, commonly from cruise reports, including water depth (in meters), recovery year, latitude and longitude, coring technology, and core length (in meters).

Summary details regarding affiliated research vessels and institutions were gathered from publications or cruise reports. Cruise reports were commonly available for research expeditions affiliated with

JAMSTEC, GEOMAR, and Scripps Institution of Oceanography, and less commonly available for older cores. All evidence used in age model development was reported, along with sedimentation rates and the sedimentary age ranges, to provide investigators with the capacity to quickly evaluate specific cores that meet the investigative priorities. For each core, paleoceanographic proxies and associated publications are documented, to provide an efficient resource for assessing the availability and quality of different lines of paleoenvironmental information. In addition, we evaluated the annual number of age models published (using any dating technique), age models published specifically with radiocarbon dating, publications generated, sediment cores collected, research cruises completed, and the mean number of proxies generated per core.

**3.0      Results**

**3.1      Sediment coring and metadata**

We documented 2134 sediment cores and 283 marine geology research cruises above 30°N, from 1951 to

2016, in the North Pacific, the Bering Sea, the Sea of Japan, and the Sea of Okhotsk (Figure 1, Table 1).

The majority of sediment cores were extracted from the Northern Subarctic Pacific (1391 cores), followed by the Sea of Japan (349 cores), the Sea of Okhotsk (271 cores), and the Bering Sea (123 cores). Many of the oldest cores in this oceanographic province came from the central abyssal Pacific and were associated with the Deep Sea Drilling Project. Cores were extracted from the North Pacific from 1951- 2010, and the oldest age models extend to 120,000 ka (Figure 1, Table 1). Metadata associated with sediment cores or marine research cruises are frequently unavailable or omitted from publications affiliated with sediment cores. For example, 495 cores are in the literature without recovery year, 354 sediment cores were published without latitude and longitude, and 642 cores were reported without specifying the coring technology used (Table 1, 2). Moreover, 1210 sediment cores reported in our database were identified in supplemental tables within publications, however no original cruise reports or peer-reviewed publications otherwise report on these cores.

**3.2      Sediment chronologies**

In the North Pacific, 519 marine sediment cores have published age models, and 266 of these chronologies are generated with radiocarbon dating ($^{14}$C) of planktonic foraminifera, molluscs, or terrigenous material like bark or wood fragments (Figure 2). Radiocarbon dating is the most common chronological technique region-wide (51% of age models incorporate this method). Lead dating ($^{210}$Pb) is used for 12 sediment chronologies. Many other lines of evidence are used in the North Pacific and marginal seas to develop paleoceanographic age models. These approaches vary regionally and include planktonic foraminifera oxygen isotope stratigraphy, diatom silica oxygen isotope stratigraphy, biostratigraphy, magnetostratigraphy, tephrochronology, chronostratigraphy, carbonate stratigraphy, opal stratigraphy, composition, lithological proxies, the presence of lamination, chlorin content, and color (a*, b*, and L* values) (Figure 2, Table 3). For example, in the Sea of Japan, lithological proxies such as core laminations are often utilized as chronological proxy evidence, and 12% of local age models incorporate this technique. Tephrochronology is also applied in 51% of Sea of Japan age models due to regional volcanism. In the Bering Sea, peaks in silica are often used, and 13% of the regional age models incorporated this technique. Published sedimentation rates ranged across the North Pacific (0.1-2000

cm/ka), Bering Sea (3-250 cm/ka), Sea of Okhotsk (0.7-250 cm/ka), and the Sea of Japan (0.2-74 cm/ka), with the highest rates within the Alaska Current in the North Pacific (up to 2000 cm/ka).

**3.3     Paleoceanographic proxies from marine sediment cores**

From all reported sediment cores in the North Pacific and marginal seas, only 40% of cores have published proxy data (Figure 3, 5). Stable isotope stratigraphy was available for 293 cores, including oxygen, carbon, or nitrogen isotopes ($\delta^{18}$O, $\delta^{13}$C, $\delta^{15}$N). We documented planktonic (236 cores) and benthic (178 cores) foraminiferal oxygen isotopes, planktonic (67 cores) and benthic (77) foraminifera carbon isotopes, and 34 cores with bulk sediment nitrogen isotopes. Of note, 98 cores were available with magnetostratigraphy (Table 4, Figure 4).

We recorded paleothermometry data for 234 cores, including planktonic foraminifera oxygen isotopes, magnesium/calcium ratios from planktonic foraminifera, and alkenones TEX$_{86}$, and U$^k_{37}$ (Figure 3). We recorded 425 cores with microfossil biostratigraphy, including foraminifera, diatom, radiolarian, ostracod, silicoflagellate, ebridian, acritarch, coccolithophore, or dinoflagellate assemblages (Figure 3, Table 4).

Biostratigraphy utilizes known microfossil assemblages and their corresponding ages to assign a geologic age range to core strata containing assemblages.  Geochemical analyses are reported for 151 cores, including measurements of, for example, brassicasterol, magnesium, calcium, molybdenum, cadmium, manganese, uranium, chromium, rhenium, chlorin, titanium, iron, zinc, and beryllium (Figure 3, Table 3).

[revised manuscript text omitted]

15: 528–536. doi:10.1029/1999pa000473.

---

## Author Comment (AC2) · 25 Aug 2017

Thank you very much for your constructive comments and suggestions. Below we comprehensively respond to each request and critique.

"This is a very useful compilation, and the authors are complimented on their hard work. It would be nice to have follow ups for other parts of the world's oceans. For this study, limitations include that only cores north of 30deg N were included."

We thank you for your comment, and agree that follow-ups for other parts of the global ocean would be useful to the paleoceanographic community. A good starting point may be the Equatorial Pacific and the Southern Ocean.

"The paper needs critical discussion of cores having benthic O18 stratigraphy and magnetostratigraphy seems missing too. These are essential for anyone seeking to identify cores useful in Quaternary paleoceanography."

We thank the reviewer for this critique. In total, there are 236 planktonic $\delta$18O records, and 178 benthic $\delta$18O records. This information is in section 3.3, line 174. We now include text explicitly discussing the 129 cores with benthic $\delta$18O and 98 cores with magnetostratigraphy. We have added this information in section 4.1, line 227, after we discuss the use of planktonic stratigraphy. We have additionally added Figure 4 to visualize the distribution of both benthic and planktonic $\delta$18O stratigraphy across the pacific.

"Which raises the question, do the authors know the oldest sediments deposited in the cores, or at least some of them."

The oldest sediments in our database are 120,000 ka. We have added information about this in line 145.

"I raised the cruise report topic too. Can they discuss briefly the quality of cruise reports for non-DSDP/ODP cores? Is there a repository for cruise reports or PDFs of older reports? These resources would be critical for cores not yet published on."

The IODP cruise reports were the gold standard for accessibility, organization, and thoroughness. Cruise reports were often available for JAMSTEC, GEOMAR, and Scripps cruises, but others were harder to find, and usually found as citations in the papers published on the cores taken during said cruises. Some were print-only and located in the University of Washington library or borrowed from the inter-library network. We extracted metadata from hand-written notes, text, or report tables. In summary, there is no common repository for all cruise reports, but in cases where dois are reported or url for the reports were published, we cited them in our database's bibliography. We have added this information into section 4.3, line 263.

"So, I suggest minor revisions. I made a few comments in the attached PDF supple-

ment. Please also note the supplement to this comment: https://www.earth-syst-sci-data-discuss.net/essd-2017-49/essd-2017-49-RC1-supplement.pdf."

Below we address line-by-line edits.

Line 35: "Is lithophysical a word? Physical properties? Lithological?"

We change the word lithophysical to lithological across the manuscript.

Line 40: "No oxygen?"

Thank you for catching this oversight. We have added oxygen to this example of our isotopic proxy evidence in line 40.

Line 105: "Or millions?"

We use ka, thousands of years ago for most sediment cores, though a few reach millions of years. We have added this distinction into line 99.

Line 132: "Please comment here about how often cruise reports are available, how you handled the un-evenness of who did a cruise report and is it accessible."

Cruise reports were available mostly for iterations of IODP cruises, as well as modern research institutions such as Scripps, JAMSTEC, and GEOMAR. Where cruise reports were unavailable, we grouped cores by common publications and the notes about the cruises included within said publications. This information is provided in section 4.3, line 263.

Line 147: "Not 2016?"

We have not documented new cores after 2010, which we speculate is a product of the delay between the coring, research, and publication process.

Line 178: "OR benthic, not AND; lots of studies lack both".

Thank you for this catch. We see this difference and since changed the format of this text in line 173.

Line 183: "Paleobiology is one word typically."

We have changed this phrase to one word in line 105.

Line 190: "Or not and"

We have changed "and" to "or" in line 182.

Line 191: "Typically an age range or a min/max age."

We have changed "date" to "age range" in line 184.

Line 195: "What about paleomagnetics, magnetostratigraphy, or MS etc., somewhere in these paragraphs?"

We have added text regarding the presence of magnetostratigraphy in Pacific paleoceanographic sequences, and have added information about it in section 3.3, line 176.

Line 356: "Define benthic delta 18 Ob = benthic, etc"

We have added a description of the benthic and planktonic subscripts to table 4 in the caption. Line 358: Table 4: "What about calc. nannofossil biostrata. More important than forams for the Quaternary."

Thank you for this. We have added regional numbers for calcareous nannofossil biostratigraphy, including coccolithophores, and ostracods, into table 4.

Line 390: "I think it is essential to show, discuss how many cores had BENTHIC oxygen isotope stratigraphy for chronology, tuning to orbital patterns, use in the LR04 and other stacks."

We agree, thank you for the note on this oversight. We have added two maps as figure 4 showing where benthic isotope stratigraphy and planktonic isotope stratigraphy are used in age models, and added text to results section 3.3 (line 175) and conclusion section 4.1 (line 227).

Please also note the supplement to this comment:
https://www.earth-syst-sci-data-discuss.net/essd-2017-49/essd-2017-49-AC2-
supplement.pdf

———————————————————

[Figure]

**Supplement:**

**A database of paleoceanographic sediment cores from the North Pacific, 1951-2016**

Marisa Borreggine[1], Sarah E. Myhre[1,2*], K. A. S. Mislan[1, 3], Curtis Deutsch[1], Catherine V. Davis[4]

[revised manuscript text omitted]

**1.1    Assembling a paleoceanographic database**

A clear need exists for high quality paleo-environmental reconstructions to fit the North Pacific into a climate global framework, because the role this enormous ocean basin plays in earth system changes remains relatively unclear in comparison to the Southern and Atlantic Oceans. To address the collective need, we present here a new database of North Pacific paleoceanographic research efforts, along with the broad findings of our census of coring metadata, age model development, and proxy publications. We address the following questions in this manuscript to supplement and provide context for our database:

1.  Where have sediment cores been extracted from the North Pacific seafloor (North of 30°N,

  including the Pacific Subarctic Gyre, Alaskan Gyre, Japan Margin, and California Margin),

  the Bering Sea, the Sea of Okhotsk, and the Sea of Japan? What metadata is available in

  published cruise reports or peer-reviewed investigations, including core name, recovery date,

  recovery vessel and scientific agency, latitude and longitude, water depth, core length, and

  coring technology?

2.  For sediment cores with published age models, what lines of evidence were used to develop

  the chronological age of the sediment, what is the age range from core top to core bottom,

  and what are the sedimentation rates?

3.  What lines of sedimentary, geochemical, isotopic, and biological proxy evidence are

  available for each sediment core?

4.  How has the state of North Pacific research efforts and reporting changed since the beginning

  of paleoceanographic expeditions?

**1.2      Paleoceanographic age models, proxies, and nomenclature**

Marine sedimentary age models tie the sedimentary depth (in meters below sediment surface) to calendar age (ka, thousands of years and/or Ma, millions of years). Not all sedimentary chronologies are of the same quality, and often age models are iteratively refined. Age models are developed with many different dating techniques, which are dependent upon the quality, preservation, and age of the sediments, as well as the investigative priorities of research teams and the proximity of other well-developed sedimentary chronologies. Paleoceanographic proxies, including biological, isotopic, geochemical, and sedimentary observations and measurements, address large thematic questions in the reconstruction of ocean environments, including ocean temperature, paleobiology, seafloor geochemistry, sea ice distribution, and additional sedimentary analyses.

Sediment cores are often represented by their cruise-core unique identifier, which has the general format

"cruise name-core number". The cruise number is generally indicative of the research vessel employed and the year of the expedition. For example, L13-81 is the 13[th] cruise of S.P. Lee in 1981, MR06-04 is the

4[th] cruise of R/V Mirai in 2006, and YK07-12 is the 12th cruise of R/V Yokosuka in 2007. Often the core number will be preceded or followed by a PC (piston core), MC (multicore), TC (trigger core), or GC

(gravity core) to signify the coring technology. The sediment core B37-04G is the 4[th] gravity core from the 37[th] cruise of R/V Professor Bogorov, and EW9504-11PC is the 11[th] piston core from the fourth cruise of R/V Maurice Ewing in 1995. However, this nomenclature is not comprehensive. For example, cores affiliated with iterations of the International Ocean Discovery Program are represented by the program abbreviation and their hole number (i.e. ODP Hole 1209A). Cruise name or number is unknown for many cores, and in these cases the core is referred to by number.

**2 Methods**

Here we assembled a database from peer-reviewed publications, publicly available online cruise reports, and print-only cruise reports available through the University of Washington library network. Detailed metadata was reported for cores where it is available, commonly from cruise reports, including water depth (in meters), recovery year, latitude and longitude, coring technology, and core length (in meters).

Summary details regarding affiliated research vessels and institutions were gathered from publications or cruise reports. Cruise reports were commonly available for research expeditions affiliated with

JAMSTEC, GEOMAR, and Scripps Institution of Oceanography, and less commonly available for older cores. All evidence used in age model development was reported, along with sedimentation rates and the sedimentary age ranges, to provide investigators with the capacity to quickly evaluate specific cores that meet the investigative priorities. For each core, paleoceanographic proxies and associated publications are documented, to provide an efficient resource for assessing the availability and quality of different lines of paleoenvironmental information. In addition, we evaluated the annual number of age models published (using any dating technique), age models published specifically with radiocarbon dating, publications generated, sediment cores collected, research cruises completed, and the mean number of proxies generated per core.

**3.0      Results**

**3.1      Sediment coring and metadata**

We documented 2134 sediment cores and 283 marine geology research cruises above 30°N, from 1951 to

2016, in the North Pacific, the Bering Sea, the Sea of Japan, and the Sea of Okhotsk (Figure 1, Table 1).

The majority of sediment cores were extracted from the Northern Subarctic Pacific (1391 cores), followed by the Sea of Japan (349 cores), the Sea of Okhotsk (271 cores), and the Bering Sea (123 cores). Many of the oldest cores in this oceanographic province came from the central abyssal Pacific and were associated with the Deep Sea Drilling Project. Cores were extracted from the North Pacific from 1951- 2010, and the oldest age models extend to 120,000 ka (Figure 1, Table 1). Metadata associated with sediment cores or marine research cruises are frequently unavailable or omitted from publications affiliated with sediment cores. For example, 495 cores are in the literature without recovery year, 354 sediment cores were published without latitude and longitude, and 642 cores were reported without specifying the coring technology used (Table 1, 2). Moreover, 1210 sediment cores reported in our database were identified in supplemental tables within publications, however no original cruise reports or peer-reviewed publications otherwise report on these cores.

**3.2      Sediment chronologies**

In the North Pacific, 519 marine sediment cores have published age models, and 266 of these chronologies are generated with radiocarbon dating ($^{14}$C) of planktonic foraminifera, molluscs, or terrigenous material like bark or wood fragments (Figure 2). Radiocarbon dating is the most common chronological technique region-wide (51% of age models incorporate this method). Lead dating ($^{210}$Pb) is used for 12 sediment chronologies. Many other lines of evidence are used in the North Pacific and marginal seas to develop paleoceanographic age models. These approaches vary regionally and include planktonic foraminifera oxygen isotope stratigraphy, diatom silica oxygen isotope stratigraphy, biostratigraphy, magnetostratigraphy, tephrochronology, chronostratigraphy, carbonate stratigraphy, opal stratigraphy, composition, lithological proxies, the presence of lamination, chlorin content, and color (a*, b*, and L* values) (Figure 2, Table 3). For example, in the Sea of Japan, lithological proxies such as core laminations are often utilized as chronological proxy evidence, and 12% of local age models incorporate this technique. Tephrochronology is also applied in 51% of Sea of Japan age models due to regional volcanism. In the Bering Sea, peaks in silica are often used, and 13% of the regional age models incorporated this technique. Published sedimentation rates ranged across the North Pacific (0.1-2000

cm/ka), Bering Sea (3-250 cm/ka), Sea of Okhotsk (0.7-250 cm/ka), and the Sea of Japan (0.2-74 cm/ka), with the highest rates within the Alaska Current in the North Pacific (up to 2000 cm/ka).

**3.3     Paleoceanographic proxies from marine sediment cores**

From all reported sediment cores in the North Pacific and marginal seas, only 40% of cores have published proxy data (Figure 3, 5). Stable isotope stratigraphy was available for 293 cores, including oxygen, carbon, or nitrogen isotopes ($\delta^{18}$O, $\delta^{13}$C, $\delta^{15}$N). We documented planktonic (236 cores) and benthic (178 cores) foraminiferal oxygen isotopes, planktonic (67 cores) and benthic (77) foraminifera carbon isotopes, and 34 cores with bulk sediment nitrogen isotopes. Of note, 98 cores were available with magnetostratigraphy (Table 4, Figure 4).

We recorded paleothermometry data for 234 cores, including planktonic foraminifera oxygen isotopes, magnesium/calcium ratios from planktonic foraminifera, and alkenones TEX$_{86}$, and U$^k_{37}$ (Figure 3). We recorded 425 cores with microfossil biostratigraphy, including foraminifera, diatom, radiolarian, ostracod, silicoflagellate, ebridian, acritarch, coccolithophore, or dinoflagellate assemblages (Figure 3, Table 4).

Biostratigraphy utilizes known microfossil assemblages and their corresponding ages to assign a geologic age range to core strata containing assemblages.  Geochemical analyses are reported for 151 cores, including measurements of, for example, brassicasterol, magnesium, calcium, molybdenum, cadmium, manganese, uranium, chromium, rhenium, chlorin, titanium, iron, zinc, and beryllium (Figure 3, Table 3).

[revised manuscript text omitted]

15: 528–536. doi:10.1029/1999pa000473.